# Novel Biomarkers of Hepatitis B Virus and Their Use in Chronic Hepatitis B Patient Management

**DOI:** 10.3390/v13060951

**Published:** 2021-05-21

**Authors:** Alicia Vachon, Carla Osiowy

**Affiliations:** 1Department of Medical Microbiology and Infectious Diseases, University of Manitoba, Winnipeg, MB R3E 0J9, Canada; vachona@myumanitoba.ca; 2National Microbiology Laboratory, Public Health Agency of Canada, Winnipeg, MB R3E 3R2, Canada

**Keywords:** hepatitis B virus, biomarker, qHBsAg, serum HBV RNA, pgRNA, quantitative anti-HBc, HBcrAg, NRAg

## Abstract

Even though an approved vaccine for hepatitis B virus (HBV) is available and widely used, over 257 million individuals worldwide are living with chronic hepatitis B (CHB) who require monitoring of treatment response, viral activity, and disease progression to reduce their risk of HBV-related liver disease. There is currently a lack of predictive markers to guide clinical management and to allow treatment cessation with reduced risk of viral reactivation. Novel HBV biomarkers are in development in an effort to improve the management of people living with CHB, to predict disease outcomes of CHB, and further understand the natural history of HBV. This review focuses on novel HBV biomarkers and their use in the clinical setting, including the description of and methodology for quantification of serum HBV RNA, hepatitis B core-related antigen (HBcrAg), quantitative hepatitis B surface antigen (qHBsAg), including ultrasensitive HBsAg detection, quantitative anti-hepatitis B core antigen (qAHBc), and detection of HBV nucleic acid-related antigen (HBV-NRAg). The utility of these biomarkers in treatment-naïve and treated CHB patients in several clinical situations is further discussed. Novel HBV biomarkers have been observed to provide critical clinical information and show promise for improving patient management and our understanding of the natural history of HBV.

## 1. Introduction

It is estimated that over 257 million people are chronically infected with hepatitis B virus (HBV) worldwide and over 880,000 annual deaths are the result of hepatitis B-related outcomes such as hepatocellular carcinoma (HCC) and liver cirrhosis [1]. Although childhood vaccination programs have been operational since the 1990s, a significant number of individuals worldwide live with the life-changing disease that is hepatitis B and therefore require monitoring of treatment response, viral activity, and disease progression to minimize their imminent risk of developing HBV-related liver disease. While qualitative detection of traditional markers such as HBV DNA, HBV e antigen (HBeAg), HBV surface antigen (HBsAg), and antibody to the HBV core antigen (AHBc) are used in monitoring of acute or chronic hepatitis B, these markers have limitations in predicting clinical outcomes of disease or antiviral treatment. Quantification of intrahepatic covalently closed circular DNA (cccDNA) is the gold standard for gaining a full understanding of HBV replicative and transcriptional activity; however, invasive procedures and a lack of standardization prevent this as a routine prognostic HBV biomarker. Quantification of novel and traditional serum HBV markers is being investigated as a surrogate of cccDNA, not only to circumvent the need for a liver biopsy to measure viral transcriptional activity, but also to provide additional information on the state of disease, allow for more refined guidance in the clinical management of hepatitis B, and improve our understanding of HBV natural history. Methods to detect and quantify novel serum markers of HBV have been developed and studied within the last two decades. These biomarkers (Table 1) include serum HBV RNA, hepatitis B core-related antigen (HBcrAg), quantified HBsAg (qHBsAg), quantified AHBc (qAHBc), and the detection of HBV nucleic acid-related antigen (HBV-NRAg). These markers have shown clinical utility in many studies, including with both treatment-naïve and treated chronic hepatitis B (CHB) patients. This clinical utility and the methods by which the novel HBV biomarkers are measured are the focus of this review.

## 2. HBV Replication, Natural History, and Antibody Response to Infection

Hepatitis B virus is a bloodborne pathogen transmitted by both parenteral (such as through intravenous drug use) and mucosal (such as through sexual contact) transmission, with mother-to-infant transmission being an important source of endemic infection in the absence of maternal screening and infant vaccination [2]. HBV causes severe liver disease, leading to liver cirrhosis and hepatocellular carcinoma. HBV binds via its surface antigen to the sodium taurocholate co-transporting polypeptide (NTCP) on the surface of hepatocytes to enter and infect cells [3]. After internalization of HBV, the viral relaxed circular DNA (rcDNA) genome is transported to the nucleus where it is completed into cccDNA by host enzymes. cccDNA is used as a template for transcription by host enzymes of four polyadenylated transcripts. A 2.4 kb transcript produces the large surface protein, a 0.7 kb transcript produces HBx and a 2.1 kb transcript produces the small and medium surface proteins [4]. Additionally, two approximately 3.5 kb transcripts, the pre-core RNA (pcRNA) and pre-genomic RNA (pgRNA), are greater than genome length. pcRNA is translated into HBeAg and pgRNA is translated into the core protein and viral polymerase [5]. Within the cell cytoplasm, pgRNA is also selectively encapsidated into a core protein-containing capsid particle together with the viral polymerase, by means of the Ɛ stem-loop secondary structure at the 5′ end of the pgRNA transcript. This cytoplasmic construct functions as the vehicle for viral genome replication and synthesis of rcDNA prior to envelopment and secretion or for recycling to the nucleus to generate further cccDNA [5]. Figure 1 provides a simplified overview of the HBV infection and replication cycle to show the pathways leading to the various HBV biomarkers of interest (Table 1). The pathway for production of AHBc is described below.

The natural history of CHB involves five phases of infection. Classification of patients into these phases involves longitudinal determination of HBeAg status, HBV DNA levels, ALT levels and presence of liver inflammation [6]. According to the European Association for the Study of the Liver 2017 guidelines [6], the first phase of chronic infection is the HBeAg-positive chronic infection phase. In this phase, HBeAg is present in the serum. This phase is characterized by high levels of HBV DNA, normal ALT levels and minimal liver inflammation, suggesting a lack of immune recognition. The second phase is the HBeAg-positive chronic hepatitis phase, in which there is presence of HBeAg, high HBV DNA, elevated ALT and moderate to severe liver necroinflammation due to progressive immune recognition. In this phase, most patients achieve HBeAg-negative status. The third phase is the HBeAg-negative chronic infection phase, characterized by the absence of HBeAg and presence of anti-HBe antibodies. In this phase, there is low to undetectable levels of HBV DNA and normal ALT. In the fourth phase, the HBeAg-negative chronic hepatitis phase, there is moderate to high levels of HBV DNA, high ALT levels and significant liver necroinflammation. The fourth phase is associated with viral mutants that modulate or prevent the expression of HBeAg. The fifth phase is the HBsAg-negative phase. While this phase is rarely achieved, it is the goal of current therapies as there is minimal risk for development of liver-related disease. This phase is characterized by the absence of HBsAg with or without the presence of anti-HBs antibodies. HBV DNA becomes undetectable and ALT levels normalize; however, cccDNA is still detectable in the liver and virological relapse is common. In the chronic infection and the HBsAg-negative phases, patients have a low risk of developing fibrosis or cirrhosis whereas patients in the chronic hepatitis phases have an increased risk of developing liver-related disease [6].

The hepatitis B-specific CD4 T cell response mainly targets the core protein, resulting in B cell stimulus and production of AHBc antibodies, but has also been reported to minimally target HBsAg, viral polymerase and HBx [7]. While all patients naturally infected with HBV develop AHBc antibodies, AHBc IgM antibodies are no longer detectable 4 to 8 months after initial infection but can reappear during flares of immune activation during chronic infection [8]. Neutralizing anti-HBs antibodies are detectable after effective vaccination and recovery from acute infection [8]. Both the innate and adaptive immune responses to HBV dictate the likelihood of developing chronic HBV in an age-dependent manner, such that >95% of adults will spontaneously resolve following acute infection, while infected infants have a >95% risk of chronicity [2,9].

## 3. Traditional HBV Diagnostic Testing

While some serological markers are useful in the diagnosis of HBV, such as HBsAg and AHBc antibody, others help identify the phase of CHB, such as HBeAg, anti-HBe, anti-HBs, and serum HBV DNA. HBeAg is a marker of active viral infection, such that seroconversion to anti-HBe antibody positivity results in inactive disease, with normalized liver enzyme levels and decreased HBV DNA levels [10]. However, viral activity and ALT fluctuations persist within the HBeAg-negative chronic hepatitis phase of infection due to genomic mutations altering HBeAg expression [11]. Quantified HBeAg levels have also been shown in the literature to be associated with HBeAg seroconversion and pegylated-interferon α (peg-IFN) and nucleos(t)ide analogue (NA) treatment response [12,13,14,15]. However, there is no commercially available quantitative method and no standardization between laboratory-developed methods. While qHBeAg is considered a novel biomarker of HBV in progress, further development and research are required to generate an assay and clarify its clinical value. The gold standard to determine the level of virus in the liver is to quantify cccDNA from liver biopsy samples, albeit liver biopsies are no longer routinely performed, as non-invasive surrogate measures of liver fibrosis are preferred. In addition to markers of HBV presence, monitoring the physiopathological state of the liver and development of HCC are crucial in the standard of care for patients infected with HBV. Notable tools for this are transient elastography, and serum biomarkers for cirrhosis or HCC such as ALT, AST, and α-fetoprotein, often in combination with risk formulas such as calculation of the AST to platelet ratio index (APRI) or the Fibrosis-4 index (FIB-4) [16,17].

## 4. HBV Treatments and Outcomes

There are two types of therapies available for the treatment of hepatitis B infection: NA and peg-IFN. NAs include lamivudine (LAM), telbivudine (LdT), tenofovir disoproxil fumarate (TDF), adefovir (ADV), and entecavir (ETV), with ETV and TDF or tenofovir alafenamide fumarate (TAF) recommended as frontline therapies for treatment-naïve patients [18]. NAs are considered effective for lessening HBV viral load in the serum but do not fully reduce HBV DNA or cccDNA in the liver [19,20,21] as NAs inhibit the reverse transcription activity of the viral polymerase [22] but do not prevent recycling of rcDNA to cccDNA. Interferon treatment decreases the transcriptional activity of cccDNA and upregulates cell immunity. Interferon-α has both antiviral and immunomodulatory activity, including induction of a multitude of interferon-stimulated genes (ISGs) such as IFN-α-inducible protein 27 [23] and APOBEC3A [24]. These ISGs inhibit HBV gene expression and DNA replication or selectively deaminate and degrade cccDNA, as in the case of IFN-α-triggered APOBEC3A. Unfortunately, interferon therapy has many significant side effects and a sustained virological response is only achieved in approximately one third of newly treated patients [25]. It must be stated that while the available therapies can be very effective at inhibiting the replication of HBV, virological and/or clinical relapse after treatment cessation are common, heightening the risk for liver disease. It is therefore crucial to identify markers that can predict relapse in patients who meet requirements for treatment cessation.

The main goal of treatment is to prevent the progression of severe liver disease to cirrhosis, liver failure and HCC through a reduction in viral replication, resulting in less liver damage and ultimately a lower risk of severe disease outcomes [18]. Current HBV treatments do not cure CHB (i.e., resulting in complete elimination of cccDNA [26]) and so the optimal outcome of treatment is functional cure. Functional cure is described as loss of serum HBsAg, undetectable serum HBV DNA and low intrahepatic persistence of HBV DNA [27]. New therapeutics targeting viral entry, transcriptional activity, and assembly are being investigated in addition to therapies targeting cccDNA and immune therapies in the hope of being able to achieve HBV cure.

## 5. Novel HBV Biomarkers

It has become increasingly clear that while current markers of HBV are informative in the monitoring of CHB, they are unsatisfactory for the prediction of clinical outcomes. Additionally, detection and measurement of liver cccDNA, considered the gold standard method for monitoring HBV replication, presents several problems. Firstly, this method requires a liver biopsy sample, which is not only invasive, but also poses several risks to the patient, such as pain, bleeding, or infection [28,29]. Secondly, cccDNA measurement methods are not standardized and include several technical obstacles, such as the possible selection of non-representative biopsy material [28,30,31]. Following successful response to antiviral treatment, serum HBV DNA is reduced to undetectable levels and ALT levels stabilize over time, rendering these markers futile for monitoring of CHB. For these reasons, novel serum markers representative of the replicative and transcriptional activity of HBV in the liver are necessary and will serve to accurately guide treatment management and clinical outcome prediction.

### 5.1. Serum HBV RNA/pgRNA

HBV pre-genomic RNA has been detected in cultured hepatocytes, patient liver biopsies and patient serum [32,33,34,35]. HBV pgRNA, a transcript derived from HBV cccDNA used for genome replication, is the main component of serum HBV RNA. pgRNA has been detected in its full length form (flRNA) [32], as well as a 3′ truncated and cryptic polyadenylated form (trRNA) [32,36], a non-polyadenylated 3′ truncated form [37] and as spliced variants of flRNA [38]. Additionally, Stadelmayer et al. have detected variants of the HBx transcript in the plasma of HBeAg-positive chronically infected individuals using 5′ RACE RT-PCR [33]. Several studies have provided evidence that HBV RNA can be used as a surrogate marker of cccDNA transcription in the liver in a variety of clinical situations.

Many studies have evaluated the correlation between HBV RNA and other traditional or novel HBV biomarkers to investigate HBV natural history and determine the utility of serum HBV RNA as an alternative biomarker. In treatment-naïve patients within multi-ethnic cohorts, it has been shown that levels of serum HBV RNA are higher in HBeAg-positive CHB patients than in those who are HBeAg-negative [22,29,39,40,41]. Additionally, a strong correlation between serum HBV DNA and serum HBV RNA levels in treated and untreated patients has been reported [22,32,40,42]. However, this correlation has not been consistently observed in treatment-naïve patients [39,43]. Serum HBV RNA was also found to correlate with HBcrAg [39] and moderately correlate with HBsAg and HBeAg quantities [40] in treatment-naïve patients. Studies involving univariate analysis of treatment-naïve patients have shown that serum HBV RNA levels are significantly associated with age, HBeAg status, HBV genotype [40,43] as well as HBV DNA, HBsAg and HBcrAg levels [43]. The presence of pre-core or basal core promotor (BCP) mutations were also found to be positively associated with serum HBV RNA levels by multivariate linear regression [40,43]. Gao et al. studied the correlation between cccDNA levels in the liver with serum HBV RNA and other HBV biomarkers to gauge transcriptional activity in relation to treatment. This study found that liver cccDNA, collected via biopsy, correlated better with serum HBV DNA than RNA and no correlation was found between cccDNA and HBsAg or HBeAg [44].

Theoretically, serum HBV RNA should be a potential predictor of cccDNA levels in the liver; however, its utility appears to be dependent on HBeAg status [34]. A study by Wang et al., investigating the correlation among phases of chronic infection in treatment-naïve patients, demonstrated that the ratios of HBV replication intermediates to serum HBV RNA differed depending on the CHB phase. Thus, HBV RNA levels differ depending on the phase of infection and these levels correlate with replication intermediates [42]. The study found that the ratio of serum HBV RNA and liver HBV DNA was highest in the HBeAg-positive chronic infection phase as compared to HBeAg-negative phases. The correlation between serum HBV RNA and HBV DNA remained after stratification into the four phases of chronic infection although the correlation with other markers weakened [42]. Similarly, Liu et al. have found that serum HBV RNA levels were significantly lower in the HBeAg-negative chronic infection phase than in the HBeAg-negative chronic hepatitis phase, indicating that HBV RNA could be used to differentiate these two phases in treatment-naïve patients [41]. It must be noted though that HBV DNA was found to be more effective than serum HBV RNA at discerning chronic infection and chronic hepatitis phases in HBeAg-negative patients [39].

In keeping with the close association of serum HBV RNA with the HBeAg-positive phase of infection, this biomarker has the potential to predict HBeAg seroconversion during NA and/or peg-IFN treatment. van Bömmel et al. have shown that patients on NA therapy that achieved HBeAg seroconversion had a stronger decline in flRNA and trRNA levels from baseline to 6 months after treatment initiation, as compared to patients who did not achieve seroconversion. It is also of note that patients who achieved seroconversion also had a stronger decline in serum HBV DNA at month 6 of treatment [32]. In a study by Luo et al., more patients on NA treatment that had negative serum HBV RNA (19/23) achieved HBeAg loss compared to those with detectable HBV RNA (19/56), and loss was achieved within a shorter median time frame (72 weeks vs. 152 weeks) [45]. A further study found that after 12 weeks of peg-IFN monotherapy or combination peg-IFN/LAM therapy, patients who had HBeAg loss had significantly lower serum HBV RNA and those who achieved HBsAg loss had a more pronounced decline in HBV RNA from baseline levels [46]. Ma et al. also investigated the association of quantitative serum HBV RNA and HBeAg seroconversion in the context of NA or peg-IFN treatment, or a combination of both [47]. While the decrease of serum HBV RNA levels during treatment was not predictive of HBeAg seroconversion, baseline serum HBcrAg levels below 8.0 log_10_ U/mL or baseline HBV RNA levels below 6.2 log_10_ IU/mL were the best factors to predict HBeAg seroconversion by AUROC. Amidst these discrepancies, larger population cohorts may be necessary to investigate the issue of predicting HBeAg seroconversion. It is also worth noting that while van Bömmel et al. [32] measured flRNA and trRNA seperately, the exact pgRNA species detected by the Ma et al. [47] method were not specified.

One of the most promising roles for serum HBV RNA is in monitoring transcriptional activity when HBV DNA is not measurable due to treatment suppression. Although serum HBV RNA levels remain detectable during antiviral treatment, a number of studies describe serum RNA reductions in response to treatment, particularly in those achieving undetectable HBV DNA [39,48]. In a study by Mak et al., among 142 NA-treated patients, 77.5% still had detectable serum HBV RNA after 96 weeks of treatment and HBV RNA levels correlated with HBcrAg and qHBsAg [39]. While HBV DNA levels decrease significantly during NA treatment, HBV RNA levels decrease at a slower rate [34]. Additionally, individuals who achieved undetectibility of HBV DNA in under 16 weeks of NA treatment had lower levels of serum HBV RNA at week 12 of therapy than those who did not [48]. Serum HBV RNA has been reported to decline rapidly in response to peg-IFN monotherapy and peg-IFN with LAM [46], while ETV-treated patients infected with HBV genotype B or C demonstrated a rapid decline in serum HBV RNA within 4 to 12 weeks of treatment, after which serum HBV RNA levels declined at a slower rate [35]. While flRNA and trRNA levels are found to be highly correlated during NA treatment, weaker correlations are observed between HBV RNA and HBV DNA, as well as qHBsAg [32]. However, serum HBV RNA has been found to correlate with histological scores, qHBsAg and liver HBV RNA but not intrahepatic cccDNA in patients treated with NAs for over 1 year, while cccDNA levels only correlated with qHBsAg [35,44]. Lin et al. have found significantly higher serum HBV RNA levels in NA-treated CHB patients and asymptomatic carriers compared to patients with liver cirrhosis and HCC [29]. Additionally, deep sequencing of liver and serum HBV RNA has shown that serum RNA is genetically representative of liver-based RNA and the transcriptional activity of cccDNA during treatment [35]. These studies highlight the clinical usefulness of serum HBV RNA during NA treatment as a surrogate marker of cccDNA transcriptional activity, but not necessarily the absolute level of hepatic cccDNA.

Further to the role serum HBV RNA may play in treatment monitoring, is its potential value for predicting viral rebound following treatment cessation. As serum HBV RNA remains detectable in patients with completely suppressed HBV viral load levels, it may be used to identify patients with low intrahepatic cccDNA transcriptional activity, which may result in a higher likelihood of relapse once treatment has stopped. In a study by Wang et al., a cohort of 33 patients receiving standard NA therapy for >3 years having undetectable HBV DNA was examined. They observed viral rebound in all patients with detectable HBV RNA at end of treatment (EOT; 21/21), whereas only 3 of 12 patients with undetectable RNA experienced rebound [34]. In several Asian or multi-ethnic cohorts, detectable or higher levels of serum HBV RNA, either together with HBV DNA or alone, were also predictive of viral and ALT rebound [41,49,50]. Contrary to this, patients within a cohort having undetectable levels of serum HBV RNA, DNA and liver cccDNA following NA treatment for a median duration of 13.4 years, still experienced viral or ALT rebound following treatment cessation [51]. It was speculated that extended NA treatment might reduce the cccDNA pool to the extent that serum HBV RNA is too low to be detected. Thus, an important factor in the observed disconnection between serum HBV RNA levels and viral rebound may be the sensitivity of the test for detecting residual serum RNA. As of currently, there is neither a standardized method for serum HBV RNA detection, nor an international quantitative RNA standard to validate laboratory-developed tests; this will continue to hinder comparison and interpretation of studies investigating the clinical utility of serum HBV RNA.

The main methods used to detect and quantify serum HBV RNA include rapid amplification of cDNA ends (RACE)-based RT-qPCR, standard RT-qPCR, and droplet digital PCR (ddPCR) [29,34,36,37]. RACE RT-PCR of 3′ polyadenylated RNA was firstly developed by Köck et al. [52], a methodology that has been widely used to detect serum pgRNA. This method employs reverse transcription from the 3′ end of pgRNA involving a primer having a 5′ non-specific anchor sequence followed by a homopolymeric T region and a short HBV target-specific sequence at the primer 3′ end. As modified by van Bömmel et al. [32], the cDNA created is amplified and detected by real-time PCR in the presence of a plasmid or RNA standard curve to allow relative quantification of serum HBV RNA in copies/mL. Specific primers have been designed to detect flRNA and trRNA. Several groups have also adapted an RT-qPCR approach [33,34], which detects total serum HBV RNA, wherein reverse transcription involves a primer binding to the 5′ end of HBV RNA. A further method of serum HBV RNA quantification utilizes RT-ddPCR [35,53]. Wang et al. firstly used this method with amplifying primers within the preC/C and polymerase coding regions [35]. Limothai et al. also used RT-ddPCR to detect serum HBV RNA and reported an improved limit of detection compared to Wang et al. [35,53]. A commercially available serum HBV RNA test is being developed by Abbott Laboratories (Abbott Park, IL) to be run on a high throughput automated analyzer (m2000 RealTime System). The test specifically amplifies the HBV X and core coding regions at opposite ends of full length pgRNA and has a reported lower limit of quantification of 1.65 log_10_ to 1.81 log_10_ U/mL, depending on the target [22]. Although there is currently no serum HBV RNA international standard, the assay was calibrated against the WHO HBV DNA standard to allow reporting in units such that 1 U RNA is equivalent to 1 IU HBV DNA.

In conclusion, although several methodologies have been developed to detect serum HBV RNA and determine the clinical relevance of this biomarker, it is difficult to compare studies due to a lack of standardized methodology and analyte target. Despite this, serum HBV RNA has been shown to be a useful biomarker in several different clinical situations. As HBV RNA may remain detectable years after the initiation of NA therapy [35,44], it has direct potential to be a useful surrogate of cccDNA transcriptional activity in the liver.

### 5.2. HBcrAg

Hepatitis B core-related antigen (HBcrAg) is represented by three separate proteins, HBeAg, HBcAg and p22cr (Figure 1), that all share a common 149 amino acid sequence, with the latter being a post-translational product similar to HBeAg but retaining the N-terminal domain [39,54]. HBcrAg levels in serum are measured using a chemiluminescent enzyme immunoassay (CLEIA) processed using the Lumipulse automated analyzer system (Fujirebio Inc.) involving a series of monoclonal antibodies specific to the different protein targets [55]. This assay has a lower limit of detection and quantification of 2 log_10_ U/mL and 3 log_10_ U/mL, respectively [56], although a more sensitive quantification (2.1 log_10_ U/mL) version of the assay (iTACT-HBcrAg) has been developed [57]. HBeAg is the majority component of HBcrAg (72%), with HBcAg and p22cr both present at approximately 10–15% of total HBcrAg composition in the serum of HBeAg-positive patients [58]. As mutations in the basal core promotor influence the expression of HBeAg and the presence of anti-HBe antibodies in the serum can also affect HBeAg levels [54], faulty interpretation of HBcrAg results may occur. In such a scenario, cccDNA levels are unaffected, yet the interpreted relationship between HBcrAg levels and HCC may be distorted [54,59]. Furthermore, it has been shown that HBcrAg does not correlate with intrahepatic cccDNA levels in human and chimpanzee serum [58,60], but rather the serum HBcAg component of HBcrAg, primarily comprising empty virions, more closely correlated with intrahepatic cccDNA [58]. However, there have been multiple other studies describing such a correlation between HBcrAg and liver cccDNA, regardless of HBeAg status and NA treatment [55,61,62,63,64,65]. HBcrAg was found to perform better than HBV DNA, pgRNA, and qHBsAg in correlating with cccDNA, and a decline in HBcrAg was associated with cccDNA decline in NA-treated patients [61,63,64,65]. It has also been found that HBcrAg maintains a significant correlation with liver cccDNA in HBV DNA-negative patients, supporting this marker as a viable replacement marker when HBV DNA is undetectable [55].

Various studies have described a significant correlation between levels of HBcrAg and phases of CHB natural history. Although no correlation was found between ALT levels and HBcrAg [61,65,66], higher serum HBcrAg levels have been associated with higher necroinflammation and fibrosis scores [65,67], HBeAg positivity [61,67,68], and cccDNA activity in the liver [67], highlighting the presence of more active viral replication in HBcrAg-positive patients. HBcrAg was also able to distinguish between active and inactive disease in HBeAg-negative treatment-naïve or on-therapy patients [66,69,70,71] with a diagnostic accuracy of 92.4% for HBcrAg compared to 67.6% for qHBsAg levels, in a retrospective multicenter analysis of treatment-naïve European patients [66]. A cutoff of 2 to 3 log_10_ U/mL has been identified to separate HBeAg-negative patients with inactive disease from those with active hepatitis [57,70,71]. Higher HBcrAg levels have been observed in the HBeAg-positive immune tolerant phase as compared to the HBeAg-positive immune clearance phase [69,70]. Furthermore, among HBeAg-negative patients, a cutoff of 4 log_10_ U/mL has been recognized to distinguish between minimal versus mild liver disease with higher cccDNA transcriptional activity [67]. HBcrAg has been evaluated for its ability to predict HBeAg seroconversion in NA-treated patients. It has been found that patients who experience HBeAg seroconversion have lower HBcrAg levels at multiple time points during treatment as compared to patients who do not seroconvert [72,73], and a more pronounced decline in HBcrAg was observed in seroconverters [72].

HBcrAg quantification offers significant risk stratification for predicting the development of cirrhosis and other liver-related complications as well as a decreased probability of HBsAg seroclearance. In a study following HBeAg-negative patients with normal ALT and an intermediate viral load (3.30–4.20 log_10_ IU/mL) for an average duration of approximately 16 years, HBcrAg levels >4 log_10_ U/mL gave a hazard ratio (HR) of 3.22 (95% CI: 1.61–6.47) for cirrhosis risk [74]. Similarly, patients who develop HCC have higher levels of HBcrAg (>2.77–5.21 log_10_ U/mL) compared to those who do not, regardless of treatment experience [63,75,76,77,78,79,80]. This association is further strengthened in the presence of persistently low HBsAg levels (<3 log_10_ IU/mL) [60,75], although the HCC predictive power of qHBsAg is not always supported during NA therapy [76,81]. Upon stratification by HBeAg status in patients receiving NA therapy, HBcrAg levels and cirrhosis status were associated with development of HCC in HBeAg-negative patients but not HBeAg-positive patients [81,82], with a baseline HBcrAg measurement >2.9 log_10_ U/mL being associated with a higher risk of developing HCC [81,83]. Cox proportional hazard analysis among HBeAg seroconverted patients on NA treatment showed baseline HBcrAg values ranging from 4.4 to 5.21 log_10_ U/mL to be independently associated with HCC development [78,80]. It has correspondingly been found that in the case of HCC recurrence after hepatectomy that recurrence-free survival rates were lower among patients with higher HBcrAg levels (>5.2 log_10_ U/mL) [63]. Evidence for HBcrAg levels to predict HBsAg loss and sustained virological response (SVR) is limited. While differences in HBcrAg levels have been observed between HBsAg seroconverters and non-seroconverters in treated patients [84], it has been reported in multiple studies that HBcrAg levels were not associated with DNA rebound or HBsAg loss in patients on NA or peg-IFN therapy [51,67,84], or that the predictive power of HBcrAg was inferior compared to qHBsAg levels [85]. Likewise, it has been reported that levels of both HBcrAg and HBsAg at baseline are associated with SVR, but that while using both levels at week 12 of therapy had the optimal negative predictive value (NPV) for SVR, the NPV was unsatisfactory to translate to a treatment stopping rule [86].

Some insights can be made into the kinetics of HBcrAg during the course of treatment, either alone or in combination with other HBV markers. In response to therapy, HBcrAg decline was not significantly different between patients taking NAs with or without peg-IFN [85]. However, HBcrAg together with HBsAg did continue to decline following HBeAg loss during combination therapy, with a more pronounced decline in HBcrAg levels at week 72 of treatment observed in patients having HBsAg loss [85]. A decline in HBcrAg levels at specific treatment time points has been associated with SVR prediction, such that at week 12 of peg-IFN mono-therapy a decline less than 0.5 log_10_ U/mL generated a NPV of 82%, which increased to 96% when combined with a similar lack of HBsAg decline [86]. Likewise, TDF-treated patients having end-of-treatment HBsAg levels <100 IU/mL had 3-year virological relapse rates of 60.4% if baseline HBcrAg levels were >4.7 log_10_ U/mL, while only 20.3% of patients having baseline levels <4.7 log_10_ U/mL experienced relapse (*p* = 0.003) [87]. In patients treated with ETV, the median annual HBcrAg decline was higher in HBeAg-positive patients and those with high serum HBV DNA levels at baseline [88]. HBcrAg correlates with serum pgRNA levels in treatment-naïve patients as well as those on NA therapy, a correlation that was maintained throughout 96 weeks of therapy [39,43]. This correlation was found to be stronger in HBeAg-positive patients [39]. In HBeAg-positive patients treated with peg-IFN, a sustained response to treatment (<2000 IU/mL HBV DNA) was associated with combined reductions in HBcrAg (>2.7 log_10_ U/mL) and serum HBV RNA (>2 log_10_ copies/mL) at week 24 [89]. These studies illustrate that monitoring a combination of HBV biomarkers should support interpretation of treatment outcomes and improve the predictive power associated with quantitative values.

A limitation of HBcrAg testing is the single platform and limited availability of the assay in North America and Europe for non-Research Use Only application; therefore, the assessment of this marker has largely been limited to Asian populations. Further availability and assessment within larger, varied populations is required.

### 5.3. HBV Nucleic Acid-Related Antigen (HBV-NRAg)

An alternative ELISA-based method (HBV NRAg; Beijing Wantai Biological, Beijing, China) has been developed to qualitatively detect the PreS1 antigen and HBcAg, with the former having shown surrogate utility for HBV cccDNA activity and HBV replication [90,91]. Furthermore, the measurement of PreS1, or L-HBsAg, using calibrated ELISA assays, has been described in several studies to be associated with detection of residual serum HBV DNA [92], differentiation of active and inactive CHB, and prediction of functional cure via L-HBsAg to M-HBsAg ratios [93,94], although L-HBsAg levels differ by HBV genotype [95] which may confound predictive interpretation in diverse populations. The combined detection of conserved amino acid regions within both antigens [96], which has been termed ‘nucleic acid-related antigen’ [97], is performed through a dual sandwich ELISA with antigen specific monoclonal antibodies [98]. The assay claims to have higher sensitivity for PreS1 detection compared to singleplex PreS1 commercially available manual ELISAs and the assay sensitivity and specificity have been reported as 86.4–95.35% and 87.86–99.9%, respectively [96,98]. Studies conducted using the developed assay have found that HBV NRAg correlated better with serum HBV DNA presence compared to HBsAg and HBeAg [96] and the correlation with HBV DNA positivity ranges from 89.5 to 100% [97,98], regardless of HBeAg status. HBV NRAg positivity was also associated with significantly higher rates of liver abnormalities (41.13% vs. 26.95%; *p* < 0.01) compared to HBV NRAg-negative patients [98]. The manufacturer proposes HBV NRAg as a substitute for relative HBV DNA quantification as the assay has the advantage of reagent affordability and ease of use, in comparison. As the assay is qualitative, the clinical utility and predictive potential of NRAg testing remains to be seen, following further assessment with diverse patient populations.

### 5.4. Quantitative HBsAg

While qualitative detection of HBsAg may be used to screen for and diagnose HBV infection, quantitative HBsAg (qHBsAg) measurement may better inform clinicians regarding response to treatment, prediction of SVR, and disease progression, among other clinical situations. Clearance of HBsAg, either spontaneously or treatment-induced, is considered a functional cure of HBV [99], although the risk of HCC continues after seroclearance [100]. In particular, the risk was increased following HBsAg loss after treatment (within a median 4.8 years follow up) [101], likely due to the increased rate of associated cirrhosis in treated patients in comparison to treatment-naïve patients that spontaneously clear HBsAg. Surface antigen levels during infection do not always correlate with viral replicative activity, as integrated HBV DNA is a source of HBsAg expression, particularly in HBeAg-negative patients [102] (Figure 1). Furthermore, mutations within the PreS1, PreS2 or HBsAg coding regions may result in altered HBsAg quantification due to secretion or expression defects [103,104]. Thus, correlating qHBsAg and viral replicative activity or differentiating qHBsAg sourced from either cccDNA transcription or integrated DNA creates further challenges in fully understanding the clinical utility of qHBsAg. Fortunately, an HBsAg WHO international standard exists [105], upon which qHBsAg assays are standardized, allowing for direct comparison and interpretation among various patient studies and findings.

Serum HBsAg levels have been shown to correlate with other markers of HBV infection. During antiviral treatment, HBsAg levels correlate with serum HBV DNA [106,107,108] and serum HBV RNA [43], although a stronger correlation is observed in HBeAg-positive patients than those who are HBeAg negative. No correlation between qHBsAg and HBV DNA was found in HBeAg-negative patients under 40 years of age, indicating that HBsAg secretion is related to age [108]. Furthermore, treatment-naïve HBeAg-negative patients have been observed to have lower quantities of serum HBsAg, associated with lower levels of replicating virus and subviral particles, reflecting the chronic infection, or inactive phase of infection [109,110,111]. Serum HBsAg levels have been shown to reflect cccDNA levels in the liver, such that HBsAg seroclearance is associated with low intrahepatic cccDNA and a reduction in cccDNA correlates with HBsAg reduction during NA treatment [44,112,113]. However, while it has been reported by several studies that cccDNA and HBsAg levels correlate during therapy, their correlation at baseline is still under debate in HBeAg-positive patients [44,61]. Similarly, the observation that serum HBV RNA, a product of cccDNA transcription, maintains a weak correlation with qHBsAg in HBeAg-negative patients may be explained by HBsAg expression from integrated HBV DNA [40,43].

Quantitative HBsAg measurement offers a means to differentiate disease status and predict treatment response, highlighting that HBsAg levels reflect disease activity in CHB. For example, higher levels of HBsAg (>4 log_10_ IU/mL) have been reported in acute hepatitis, with a rapid decrease in the recovery phase, as compared to patients with CHB (<4 log_10_ IU/mL in 90% of patients) [114]. Low pre-treatment HBsAg levels and a strong decline of HBsAg during early treatment were found to be associated with higher rates of SVR to peg-IFN with or without NA treatment and HBsAg loss [89,113,115,116,117,118,119]. The likelihood of treatment-induced HBsAg clearance may be predicted using qHBsAg levels at baseline, such that HBeAg-negative CHB patients who achieved HBsAg seroclearance had significantly lower baseline serum qHBsAg (0.48–1.1 log_10_ IU/mL) [110,120], and baseline levels below 50 IU/mL accurately predicted HBsAg loss [110]. HBsAg levels <70 IU/mL at week 8 of treatment were also found to distinguish patients who achieved HBsAg loss from those who did not [120]. Interestingly, the composition of different surface antigen proteins in HBeAg-positive NA-treated patients could also predict HBsAg loss. Using an assay designed to quantify S-HBsAg, M-HBsAg and L-HBsAg, it was shown that patients who achieved HBsAg seroclearance had significantly lower baseline M-HBsAg levels and rapidly declining M-HBsAg and L-HBsAg concentrations, with M-HBsAg becoming undetectable at month 6 of treatment [94].

Quantitative HBsAg has also been used to predict treatment response in HBeAg-positive and -negative patients treated with peg-IFN with or without NA. Low pre-treatment qHBsAg and a strong HBsAg decline early during peg-IFN treatment were found to be associated with higher rates of SVR and HBsAg loss [89,113,115,116,117,118,119]. These observations demonstrate that HBsAg levels are a good surrogate for immune control of HBV during therapy. While no significant correlation was found between qHBsAg and HBV DNA flares during rebound in patients following cessation of long term NA therapy [51], increased qHBsAg has been observed in patients coinciding with replicative flares of HBV [114]. Replicative and biochemical flares leading to HBsAg loss have been observed in many HBeAg-negative patients stopping NA therapy and are thought to be beneficial in disease remission after treatment cessation [121,122,123,124]. It is thought that long-term therapy modulates the immune system to eliminate HBV-infected hepatocytes due to the re-expression of HBV proteins after treatment cessation [124]. Further, cytokines TNF, IL-10, IL-12p70 and IP-10 have been observed to increase after stopping NA therapy [123], indicating the activation of pro-inflammatory pathways for clearance of HBV antigens.

HBsAg levels have also been investigated as a predictor of chronic disease progression to fibrosis and HCC. Baseline levels of HBsAg (<3 log_10_ IU/mL) and HBV DNA (<3.3 log_10_ IU/mL) were associated with a minimal risk of developing HCC after 10–15 years of follow up, with qHBsAg levels as an independent risk factor in HBeAg-negative, treatment-naïve patients, while HBV DNA and ALT levels were better predictors in HBeAg-positive patients [125]. Although the correlation between qHBsAg and fibrosis stage is still a matter of debate in treatment-naïve patients [126,127], qHBsAg has been found to be a marker of HBV-specific T cell and B cell response. While HBsAg levels were reported to have no impact on the specific immune cell composition, higher HBsAg levels were associated with an exhausted phenotype in CD4+ T cells and with dysfunctional B cells [128]. These studies illustrate that qHBsAg provides direction as to the degree of HBV immune control in CHB.

Quantification of HBsAg from serum or plasma samples is normally performed using an ECLIA test performed with a high throughput automated immunoanalyzer. Without dilution, the Architect HBsAg QT (Abbott Diagnostics; Mississauga, ON, Canada) assay has a dynamic range of 0.05 to 250 IU/mL [129], while the Elecsys HBsAg II assay (Roche Diagnostics; Laval, QC, Canada) assay and the LIAISON-XL Murex HBsAg Quant (DiaSorin Canada, Mississauga, ON, Canada) assays have dynamic ranges of 0.05 to 130 IU/mL and 0.03 to 150 IU/mL, respectively. Assay platforms often include an onboard sample dilution to increase the linear range many log-fold. All three assays have been shown to correlate with each other [106]. Recently, ultrasensitive qualitative [130,131] and quantitative [132,133] HBsAg assays have been developed which reduce the limit of detection 10- to 250-fold to approximately 0.2 to 5 mIU/mL. These assays improve detection in the early window period of acute infection [130,134] and in monitoring of patient reactivation or residual, subclinical levels of HBsAg following spontaneous or treatment-associated seroclearance [135,136,137,138].

### 5.5. Quantitative Anti-HBc Antibody (qAHBc)

Antibody to the HBV core protein (AHBc) is a diagnostic marker indicating past exposure or current infection. Recently, several commercially available ELISA-based tests (anti-HBc ELISA, Beijing Wantai Biological, Beijing, China; Lumipulse G HBcAb-N, Fujirebio, Japan [139]) have been developed to allow quantification of AHBc in serum or plasma. An international reference standard has been prepared for AHBc in IU/mL unitage, allowing standardization and comparison among the various quantitative assays [140]. The Wantai ELISA has been widely used; however, it is limited by a narrow dynamic range requiring pre-dilution of specimen, distribution through a single supplier, and the need for further validation in more extensive populations [56].

As the humoral response to HBV is primarily directed towards HBcAg and this response closely follows the phases of infection [141], AHBc measurement shows efficacy for monitoring the natural history of HBV infection and the severity of liver disease. For example, AHBc levels were found to positively correlate with ALT levels (r = 0.559–0.663; *p* < 0.001), but not HBV viral load [142,143]. Similarly, regardless of HBeAg status in treated and treatment-naïve patients, lower AHBc levels (<4.62 IU/mL) were associated with low to moderate necroinflammation and fibrosis in the liver [142,144,145,146]. However, NA-treated HBeAg-positive patients were found to have significantly lower AHBc levels than HBeAg-negative patients [147], likely reflecting the increased immune activation during HBeAg-negative chronic hepatitis. Similarly, it was demonstrated that patients with resolved HBV infection receiving immunosuppressive therapy had a high risk of HBV reactivation if qAHBc levels were high and anti-HBs levels were low at baseline, with an HR of 17.29 [148]. While AHBc levels correlated with the physiological state of the liver in several studies, it was determined that AHBc levels alone are unsatisfactory for monitoring hepatic pathological stages and grades, but in combination with other quantified markers, such as HBV DNA and HBsAg, predictive interpretation is improved [149]. AHBc IgM levels were also observed to correlate with liver damage but not with HBV replication in a cohort of treatment-naïve and peg-IFN-treated patients [150]. The function of AHBc in predicting occult HBV infection has been well studied. While it has long been recognized that AHBc detection in the absence of HBsAg is often associated with occult HBV infection [151,152], qAHBc levels are better able to differentiate occult hepatitis from past exposure as well as the phases of HBsAg-positive infection [143]. For example, AHBc IgG levels are over 100-fold higher in patients experiencing overt HBV infection compared to occult infection [153]. Differentiation of the phases of CHB infection was shown by significantly higher mean AHBc levels in hepatitis phases (4.23 log_10_ IU/mL) as compared to infection phases (3.1 log_10_ IU/mL) [143,153], indicating that patients with higher immune activation against HBV have higher qAHBc levels compared to patients with inactive disease. Additionally, relative AHBc levels have been used in the past to differentiate acute and chronic infection [114,150] and more recently, to predict spontaneous HBsAg seroclearance in HBeAg seronegative patients with an AUROC of 82% within 10 years of follow up [154]. AHBc levels may also indicate the presence or absence of cccDNA in donated livers, with a negative predictive value of 94.6% [155].

Further utility of qAHBc to predict treatment response and the likelihood of relapse following treatment cessation has been demonstrated in several studies. Higher baseline qAHBc in either NA or peg-IFN-treated HBeAg-positive patients was associated with a greater HBeAg seroconversion rate (AUROC of 71–81%) [156], with optimal cut offs of 4.46 log_10_ IU/mL for patients on NA, 3.95 log_10_ IU/mL for patients on peg-IFN and 4.4 log_10_ IU/mL for combined treatment observed [156,157]. Furthermore, a progressive reduction in qAHBc levels during long-term therapy (60 months) in HBeAg-negative patients with active hepatitis infection, was shown to correlate with intrahepatic cccDNA [153], suggesting qAHBc levels as a surrogate for transcriptional activity. Similar to the practical use of qHBsAg levels to predict relapse following treatment cessation, high levels of qAHBc (≥3 log_10_ IU/mL) at EOT were associated with a lower risk of clinical relapse in patients stopping NA therapy [122,158], with improved predictive power in combination with EOT qHBsAg levels (<2 log_10_ IU/mL) [122]. Off-treatment kinetics of qAHBc were moderately associated with SVR, such that responding patients maintained stable qAHBc while relapsing patients experienced a strong increase in qAHBc levels [122]. Quantitative AHBc is particularly suited for monitoring relapse following peg-IFN treatment as it closely reflects the status of HBV-specific immunity [156]. For example, higher baseline qAHBc levels (3.14 ± 0.7 vs. 2.69 ± 0.6 log_10_ IU/mL) and increased loss of qAHBc by EOT (2.69 ± 0.6 vs. 1.95 ±0.5 log_10_ IU/mL) with peg-IFN treatment were associated with successful therapy stoppage without relapse [159].

## 6. Conclusions

As described, the collective and dynamic measurement and analysis of multiple HBV biomarkers provide a more holistic description of CHB disease activity, as no single HBV biomarker will likely answer all predictive and clinical questions. Future studies with more diverse and larger patient populations should reconcile the few discrepancies observed among various biomarker correlations and help solidify treatment stopping rules and predictive cutoff values. Measurement and analysis of novel HBV biomarkers will be complemented by the array of host markers that are being investigated to better monitor disease phase, including host genetic markers, micro-RNAs, as well as serum inflammatory and immune markers and cytokines [160,161,162]. These markers can be non-invasively measured and so extend the suite of tools for CHB disease activity interpretation. Genetic markers (single nucleotide polymorphisms) have been associated with the risk of developing cirrhosis, fibrosis or HCC [160], while host miRNA regulation can indicate the development of fibrosis or HCC [161,163]. Additionally, certain cytokines have been observed to correlate with severity of HBV disease and clinical outcomes [162,164,165,166,167]. Finally, the phenotype and functionality of HBV-specific CD8+ T cells and NK cells have been investigated for predicting HBsAg seroclearance [162]. Using both host and viral biomarkers together promises to enhance CHB patient management as host markers help define the pathophysiology of the liver and immune state of the patient while viral markers assess the replication level of HBV.

Additionally, as pointed out by Mak et al. [39], the role of novel HBV biomarkers will evolve with the advent of new treatments for chronic hepatitis B. These treatments are currently in various phases of clinical trials, and include entry inhibitors, core protein allosteric modulators (CpAMs), RNA interference therapies, cccDNA inhibitors and immune therapies. Monitoring of response to new therapies will likely be best served by incorporating both host immune markers, to monitor the efficacy of immune therapies, such as Toll-like receptor agonists, engineered T cells and therapeutic vaccines [168], and novel HBV markers, to inform the efficacy and target of novel therapies within the HBV life cycle. Recent studies investigating novel HBV therapeutics have analyzed HBV DNA, serum HBV RNA, qHBsAg, qHBeAg and HBcrAg to measure the effects of CpAMs and virus neutralizing antibody treatment on HBsAg, HBeAg and cccDNA levels in order to predict if favorable treatment outcomes may occur after treatment [169,170,171]. It is conceivable that a ‘One-Health’ approach to HBV management and treatment may arise, in which measurable viral, host and environmental targets are incorporated into a new type of precision medicine. To continue on this path, further studies will help to understand the utility and predictive power of novel HBV biomarkers as they are used more frequently in diagnostics, research and therapeutics development.

## Figures and Tables

**Figure 1 viruses-13-00951-f001:**
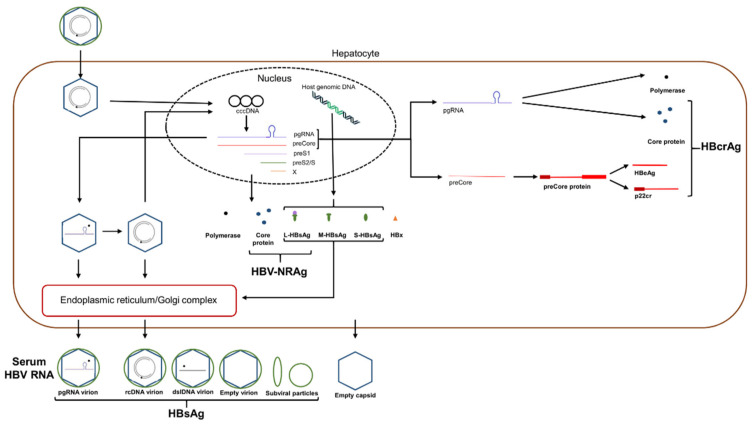
Overview of the HBV infection and replication cycle. Infecting virion particles are transported to the hepatocyte nucleus to release the relaxed circular partially double-stranded DNA genome, associated with the viral polymerase, into the nucleus. The genome is repaired and converted into cccDNA, which is the template for viral mRNA synthesis. The viral transcripts are translated following transport to the cytoplasm. The three surface antigen proteins are membrane specific and compose the viral envelope in conjunction with host lipid. Within the cytoplasm, pregenomic RNA (pgRNA) is encapsidated together with newly expressed polymerase protein within spontaneously formed capsid particles composed of core protein, to form the viral replication complex. pgRNA also serves as the template for the expression of the polymerase and core proteins. The HBV biomarkers discussed in this review, other than qAHBc, which is produced following the humoral immune response to core antigen, are shown within their expression pathways and their component parts. Note that HBsAg is detectable from replicative and non-replicative virion particles, as well as subviral particles and HBsAg expressed from HBV DNA integrated into host genomic DNA.

**Table 1 viruses-13-00951-t001:** Summary of Novel HBV Biomarkers and Available Laboratory Tests.

HBV Biomarker	Summary of Uses and Limitations	Laboratory Tests Available
Serum HBV RNA	- Surrogate of cccDNA transcriptional activity - Serum HBV RNA levels can be used to differentiate HBeAg-negative phases of CHB - Prediction of HBeAg seroconversion - Prediction of SVR in patients on peg-IFN and/or NAs - Larger cohorts and improved sensitivity required - No standardized assay to date	- In-house developed RACE-based RT-qPCR, standard RT-qPCR, and ddPCR - Abbott Diagnostics HBV RNA test (in development)
HBcrAg	- Correlates with cccDNA - Levels can distinguish between active and inactive disease - Prediction of HBeAg or HBsAg seroconversion, development of cirrhosis - Monitoring of response to treatment - Many factors may lead to faulty interpretation (i.e., anti-HBe, mutations affecting expression of HBeAg) - Limited availability of assay - Should be further investigated with larger and more diverse cohorts	- Fujirebio, Inc. CLEIA HBcrAg assay - Fujirebio, Inc. iTACT-HBcrAg
HBV-NRAg	- Higher sensitivity than PreS1 manual ELISAs - Highly correlates with HBV DNA positivity - Positivity associated with liver abnormalities - Qualitative test - Clinical utility remains to be further evaluated with larger cohorts	- Wantai NRAg manual ELISA
qHBsAg	- Monitoring of response to treatment and SVR - Prediction of HBsAg loss - Prediction of development of liver fibrosis and HCC - May reflect cccDNA levels in the liver	- Abbott Diagnostics Architect HBsAg QT - Roche Diagnostics Elecsys HBsAg II - DiaSorin LIAISON-XL Murex HBsAg Quant - NIBSC standard available (IU/mL)
Ultrasensitive HBsAg	- Limit of detection 0.2–5 mIU/mL - Early detection of acute infection - To monitor for reactivation or subclinical levels of HBsAg following seroclearance	Qualitative: - Abbott Diagnostics HBsAg Next Qualitative - Abbott Diagnostics HBsAg Next Confirmatory assay Quantitative: - Fujirebio, Inc. Lumipulse HBsAg-HQ - Sysmex Co., HBsAg ICT-CLEIA
qAHBc	- Correlates with ALT during treatment - Prediction of development of liver fibrosis and necro-inflammation - Monitoring treatment and SVR - Identification of occult HBV - Wantai assay has narrow range of quantification	- Wantai qAHBc manual ELISA (total AHBc) - Fujirebio, Inc. Lumipulse^®^ G HBcAb-N (AHBc IgG) - NIBSC standard available (IU/mL)

## Data Availability

Not applicable.

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
