# Peer review of "Novel Biomarkers of Hepatitis B Virus and Their Use in Chronic Hepatitis B Patient Management"

_viruses, 2021, doi:10.3390/v13060951_

Round 1
Reviewer 1 Report
In this manuscript, Vachon and Osiowy reviewed the new HBV biomarkers and their use in clinics. The manuscript is clear and well written. This is a very comprehensive review that will be highly useful to the field.
Specific minor comments.
- Line 85-86. It would be nice to insist a bit more on the efficacy of the immune response against HBV in adults and mention that around 95% of adults will spontaneously clear the virus during the acute phase.
- Line 116: IFNa has a broader antiviral activity than targeting cccDNA transcription. It affects many steps of the viral life cycle including cccDNA levels itself (see Lucifora et al., Science 2014).
- Typos/minor corrections:
- Line 62: I guess the authors mean "parenteral" instead of "parental"? However and independently, mother-to-child transmission of HBV would deserve to be mentioned here.
- Line 96: please defined NA here (and not in line 110).
- Line 164-165-166. The sentence "However, [...]" could be simplified.
- Line 257: replace "disconnect" by "disconnection"
- Line 259: replace "no standardized method" by "neither a starndardized method"
- Line 380: there is an extra space between "patients" and "as".
Author Response
Reviewer #1:
- Thank you for pointing out this omission. The following sentence has been added to the revised manuscript (lines 136-139): “Both the innate and adaptive immune responses to HBV dictate the likelihood of developing chronic HBV in an age-dependent manner, such that >95% of adults will spontaneously resolve following acute infection, while infected infants have a >95% risk of chronicity [2,9].”
- Thank you for pointing out this omission as well. The following sentence has been added to the revised manuscript (lines 175-179): “Interferon-α has both antiviral and immunomodulatory activity, including induction of a multitude of interferon-stimulated genes (ISG) such as IFN-α-inducible protein 27 [23] and APOBEC3A [24]. These ISG inhibit HBV gene expression and DNA replication or selectively deaminate and degrade cccDNA, as in the case of IFN-α-triggered APOBEC3A.”
- We’re very appreciative of the reviewer for catching these errors:
- Lines 62, 64-65: The word parenteral has been corrected and the sentence altered to include mother-to-child transmission (“Hepatitis B virus is a bloodborne pathogen transmitted by both parenteral (such as through intravenous drug use) and mucosal (such as through sexual contact) transmission, with mother-to-infant transmission an important source of endemic infection in the absence of maternal screening and infant vaccination [2].”).
- Lines 148-149: This is now the first definition of peg-IFN and NA.
- Lines 226-227: The sentence has been simplified as follows: “However, this correlation has not been consistently observed in treatment-naïve patients [39,44].”
- Line 326: The word disconnect has been changed to disconnection as recommended.
- Line 328: The statement has been changed to ‘neither a standardized method’ as recommended.
- Line 452: The extra space has been removed.
Reviewer 2 Report
Authors in this review provide an extensive and updated overview of novel HBV biomarkers. They critically present and discuss the clinical and research use of HBV-RNA, HBV core related antigen (HBcrAg), quantitative hepatitis B surface antigen (qHBsAg), quantitative anti-hepatitis B core antigen (qAHBc), HBV Nucleic Acid-Related antigen (HBV-NRAg). These novel HBV biomarkers together with genetic and immune host markers will be useful tools for monitoring the efficacy of actual and next future antiviral HBV therapy and to better understand HBV infection and replication. The review is well done. However, because studies reported by authors frequently refer to the different phases of infection and disease of HBV, I suggest to insert a short description of the five phases of chronic HBV infection as indicated by the EASL 2017 Clinical Practice Guidelines
Author Response
Reviewer #2:
- We thank the reviewer for their comments and suggestions to improve the manuscript. A section on the 5 phases of chronic infection according to the EASL 2017 guidelines has been added (lines 106-128).
Reviewer 3 Report
I red with great interest the manuscript by Vachon and Osiowy recapitulating novel HBV biomarkers for the management of patients with chronic HBV infection. The manuscript is well written and provides a comprehensive overview on the studies performend mainly in the last decade on this topic. In my opinion, the manuscript will be of great interest to the readers.
Below some minor comments.
1) The manuscript will benefit from a figure describing the pathways within HBV life cycle leading to the synthesis of the different biomarkers. This iconographic element will represent an added value to the present manuscript. For instance see: Mak LY et al. Gu and Liver 2019.
2) 5.2. HBcrAg. Authors reported a limit of detection of 3.0 Log U/mL for HBcrAg. However, the LLoD is 2.0 Log U/mL, while 3 Log corresponds to the limit of quantitation.
3) 5.5. Quantitative anti-HBc antibody (qAHBc). Authors should add 2 important references. 1) Caviglia GP et al. J Hepatol 2018, where it has been reported an assocciation between anti-HBc IgG levels and the presence of intrahepatic HBV cccDNA in HBsAg-neg/anti-HBC-pos liver donors, and 2) Yang HC J Hepatol 2018, where the authors observed and association between anti-HBc levels and the risk of HBV reactivation following immunesuppression in HBsAg-neg/anti-HBc-pos subjects. These two papers simultaneously published, are consistent, and highlight the the clinical value of the measurement of anti-HBc antibodies.
Anyhow, I congratulate with the authors for their manuscript.
Author Response
Reviewer #3:
- Thank you for this suggestion. We have prepared a figure as suggested. It is found at the top of page 4 with the figure heading at lines 89-103.
- We thank the reviewer for alerting us to this error. The sentence has been changed to reflect this (lines 369-370): “This assay has a lower limit of detection and quantification of 2 log10 U/mL and 3 log10 U/mL, respectively [57]…”.
- We thank the reviewer for pointing out these omissions. Several sentences have been added to correct this (lines 602-605): “Similarly, it was demonstrated that patients with resolved HBV infection receiving immunosuppressive therapy had a high risk of HBV reactivation if qAHBc levels were high and anti-HBs levels were low at baseline, with a hazard ratio of 17.29 [152].” , (and lines 623-624): “AHBc levels may also indicate the presence or absence of cccDNA in donated livers, with a negative predictive value of 94.6% [159].”